# Pathways and processes to adopting or switching to a single-dose HPV vaccination schedule in low- and middle-income countries: a qualitative study

**Sunny Roy**[1], **Megan D. Wysong**[1], **Erica N. Rosser**[1], **Ishani Sheth**[1], **Casey Geddes**[1], **Rupali J. Limaye**[1,2,3], **Joseph G. Rosen**[1]*

1 Department of International Health, Bloomberg School of Public Health, Johns Hopkins University, Baltimore, Maryland, United States of America, 2 Department of Epidemiology, Bloomberg School of Public Health, Johns Hopkins University, Baltimore, Maryland, United States of America, 3 Department of Health, Behavior, and Society, Bloomberg School of Public Health, Johns Hopkins University, Baltimore, Maryland, United States of America

* jrosen72@jhu.edu

## Abstract

The World Health Organization (WHO) recently endorsed a single-dose human papillomavirus vaccination (HPVV) schedule, but adoption of the revised immunization schedule has not been uniform, notably in low- and middle-income countries (LMICs) with outsized cervical cancer burdens. We sought to characterize the sequence of events, as well as actors/institutions involved, that facilitate or impede country-level decision-making related to the HPVV schedule. From September 2023 to February 2024, we conducted 66 semi-structured interviews with national immunization stakeholders from 19 LMICs in Africa and Asia, sampled purposively by the maturity of the HPVV program and HPVV schedule. Using purposive text analysis to synthesize emerging insights from interviews, we qualitatively discerned a sequence of decision-making pathways and processes, including actors involved at each stage of the decision-making continuum, related to adoption or switch to an HPVV single-dose schedule. We identified six key sequential steps LMICs follow when deciding on an HPV dosing schedule. Firstly, the new recommendation emerges where WHO and the Strategic Advisory Group of Experts (SAGE) release new HPV vaccine recommendations. Second, the information is shared with the Ministry of Health (MoH) and Expanded Programme for Immunization (EPI). The MoH is informed, and they convene a National Immunization Technical Advisory Group (NITAG) to assess the new HPV vaccine schedule recommendation. Third, there is the review and consideration of evidence. The NITAG reviews evidence on factors like cost-effectiveness, supply chain, disease burden, and anticipated vaccine hesitancy. Fourth, the NITAG deliberates, achieves consensus, and submits a recommendation to the MoH, which endorses the NITAG's recommendation or requests additional evidence/information before rendering a final decision on the immunization schedule. Fifth and finally, the Interagency Coordination Committee (ICC) reviews the MoH's recommendation and provides final authorization before changes to the HPV immunization schedule is communicated to Gavi.

**Data availability statement:** Data underlying the present study are generated from semi-structured, in-depth interviews with national immunization stakeholders across 19 low- and middle-income countries. Given the sensitivity of this information, and to protect participant confidentiality, data are not publicly available. However, interested parties can request de-identified transcripts by contacting the Johns Hopkins University Bloomberg School of Public Health Institutional Review Board (BSPH.irboffice@jhu.edu); all requests for access to the underlying study data will be reviewed and approved upon reasonable request.

**Funding:** This study was funded by the HAPPI Consortium via the Bill and Melinda Gates Foundation (INV-046461, INV-057603). JGR acknowledges funding from the National Institute of Mental Health (R25MH083620). The manuscript's contents are the responsibility of the authors and do not necessarily represent the official views of the funders. The funders had no role in the study design, data collection and analysis, decision to publish, or preparation of the manuscript.

**Competing interests:** The authors have declared that no competing interests exist.

## Background

Cervical cancer is among the most common cancers in women globally, with over 660,000 new cases and 348,000 deaths occurring in 2022 [1,2]. Low- and middle-income countries (LMICs) shoulder a disproportionate burden of the global cervical cancer incidence and mortality, attributable in large part to suboptimal cervical screening and treatment service provision, unavailability of preventative immunization, and socio-structural barriers to prevention and care (e.g., gender inequities, poverty) [3]. Coupled with screening and treatment, human papillomavirus vaccination (HPVV) is the cornerstone of the World Health Organization's (WHO) cervical cancer elimination strategy [4]. Since 2012, there has been substantial progress in expanding HPVV programmes in LMICs, with 32 countries receiving financial support through Gavi, the Vaccine Alliance, to introduce HPVV into their national immunization programmes—resulting in an additional 16.3 million girls in LMICs being immunized against the leading cause of cervical cancer [5].

In December 2022, the World Health Organization (WHO) and its Strategic Advisory Group of Experts of Immunization (SAGE) revised HPV immunization guidelines for permissive use of a single-dose HPVV schedule [6]. Emerging evidence from several randomized controlled trials and observational studies of single-dose HPVV efficacy and effectiveness conducted in Costa Rica [7], India [8], Kenya [9,10], and Tanzania [11] demonstrated non-inferiority of mono-dose HPVV (relative to multi-dose HPVV) in preventing persistent infection with high-risk human papillomavirus serotypes 16 and 18—responsible for over 70% of cervical cancer cases globally [12]—for at least 10 years post-vaccination. This emerging evidence of substantial immunogenicity (antibody response) and protective efficacy induced by single-dose HPVV prompted the WHO to revisit their HPVV schedule recommendations, resulting in amended guidelines that included permissive use of a single-dose regimen for non-immunocompromised girls aged 9-14 years [6]—the primary target population of HPVV efforts in LMICs receiving Gavi support.

A dose-reduction recommendation to mono-dose HPVV could have significant programmatic benefits by reducing costs associated with vaccine procurement and facilitating increased vaccination coverage through elimination of immunization tracing/follow-up efforts [13–15]. Given the novelty of the permissive single-dose recommendation, there remains limited evidence of country-level processes and pathways to adopting (or not adopting) a single-dose HPVV schedule. A seminal qualitative study of decision-making related to HPVV national introductions in LMICs described the heterogeneity of stakeholders involved in country-level dialogues related to HPVV introductions and the contributions of discrete stakeholders to different components of HPVV introduction decision-making [16]. These identified processes and influential actors, however, may materialize differently in the context of HPVV schedule decision-making, especially for LMICs that have already introduced or committed to introducing HPVV in the future.

Given this gap in understanding, we conducted a qualitative study, interviewing national immunization stakeholders from 19 LMICs at various stages of national HPVV introductions and schedule decision-making, to better characterize potential pathways to rendering decisions on the HPVV schedule, as well as underlying contextual forces that may explain variations in HPVV schedule decision-making.

## Methods

### Informants and procedures

Between September 2023 and February 2024, we leveraged professional HPVV networks (i.e., the HAPPI Consortium, the Coalition to Strengthen the Human Papillomavirus

Immunization Community, the Gavi HPVV sub-team) to systematically identify national immunization stakeholders from LMICs in Africa and Asia, purposely sampled based on the maturity of their HPVV programmes (i.e., introduced HPVV before *versus* after the WHO's revised HPVV schedule recommendation) and status of their HPVV schedule decision-making (i.e., adopted a single-dose schedule *versus* maintained a two-dose schedule and/or remain undecided about the HPVV schedule). To achieve variation in perspectives on the national HPVV single-dose decision-making process, we purposely selected immunization stakeholders representing a heterogeneity of professional affiliations within each country. Stakeholders purposefully identified and approached to participate in the study included individuals affiliated with Ministries of Health (MoH), Expanded Programmes for Immunization (EPI), National Immunization Technical Advisory Groups (NITAGs), or related immunization technical working groups (TWGs); in-country immunization technical assistance partners supporting HPVV introduction and scale-up efforts with Gavi support; civil society organizations at the intersection of HPVV and cervical cancer control; and multilateral agencies, specifically United Nations bodies.

We identified 4 discrete archetypes characterizing country progress along the HPVV introduction and schedule decision-making continua (see Fig 1). We sought to recruit national immunization stakeholders from up to 5 countries per archetype (see Table 1), intentionally sampling countries based on variations in country characteristics (e.g., geographic region, World Bank income status [17], cervical cancer incidence and mortality [2]). Within each country, we sought to interview at least 3 stakeholders for consistency and confirmability

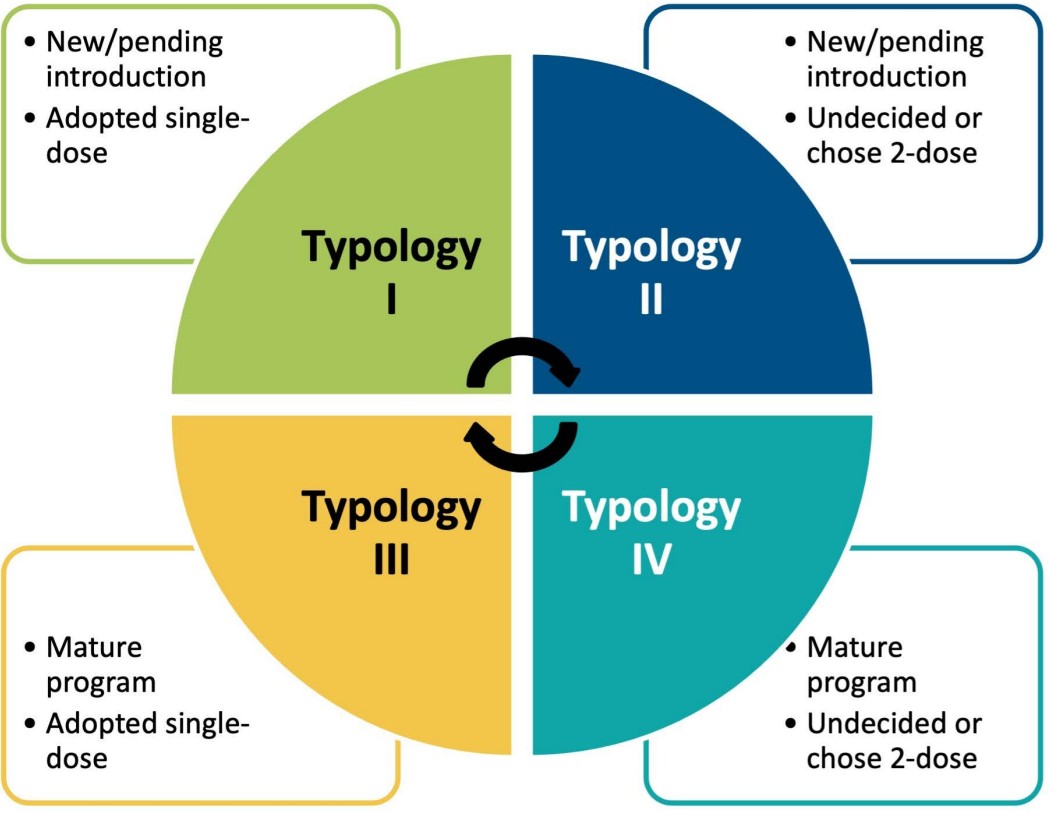

**Fig 1.** **HPV vaccination introduction and schedule decision-making archetypes from which countries were purposively sampled for study inclusion.**

**Table 1. Characteristics of countries purposively sampled for inclusion (N = 19).**

| Country | Region | Income Status | HPVV Schedule | Year of HPVV Introduction | Cervical Cancer Incidence | Cervical Cancer Mortality |
|---|---|---|---|---|---|---|
| Bangladesh | Central and South Asia | Lower-middle | Single-Dose | 2023 | 11.3 per 100,000 | 7.0 per 100,000 |
| Burkina Faso | West and Central Africa | Low | Single-Dose | 2022 | 15.9 per 100,000 | 13.0 per 100,000 |
| Cambodia | Southeast Asia | Lower-middle | Single-Dose | 2023 | 15.2 per 100,000 | 8.1 per 100,000 |
| Cameroon | West and Central Africa | Lower-middle | Single-Dose | 2020 | 33.1 per 100,000 | 25.7 per 100,000 |
| Cote d'Ivoire | West and Central Africa | Lower-middle | Single-Dose | 2019 | 32.0 per 100,000 | 20.4 per 100,000 |
| Eswatini | East and Southern Africa | Lower-middle | Two-Dose | 2015 | 95.9 100,000 | 64.3 per 100,000 |
| Ethiopia | East and Southern Africa | Low | Single-Dose | 2018 | 22.3 per 100,000 | 16.8 per 100,000 |
| Ghana | West and Central Africa | Lower-middle | Two-Dose | *N/A* | 27.0 per 100,000 | 16.9 per 100,000 |
| India | Central and South Asia | Lower-middle | Single-Dose | *N/A* | 17.7 per 100,000 | 11.2 per 100,000 |
| Indonesia | Southeast Asia | Upper-middle | Two-Dose | 2023 | 23.3 per 100,000 | 13.2 per 100,000 |
| Kenya | East and Southern Africa | Lower-middle | Two-Dose | 2019 | 32.8 per 100,000 | 21.4 per 100,000 |
| Laos | Southeast Asia | Lower-middle | Single-Dose | 2020 | 12.0 per 100,000 | 6.5 per 100,000 |
| Madagascar | East and Southern Africa | Low | Two-Dose | *N/A* | 41.8 per 100,000 | 30.0 per 100,000 |
| Nepal | Central and South Asia | Lower-middle | Two-Dose | *N/A* | 14.2 per 100,000 | 8.7 per 100,00 |
| Nigeria | West and Central Africa | Lower-middle | Single-Dose | 2023 | 26.2 per 100,000 | 14.3 per 100,000 |
| Pakistan | Central and South Asia | Lower-middle | Two-Dose | *N/A* | 5.4 per 100,000 | 3.6 per 100,000 |
| Philippines | Southeast Asia | Lower-middle | Two-Dose | 2016 | 15.5 per 100,000 | 8.0 per 100,000 |
| Tanzania | East and Southern Africa | Lower-middle | Single-Dose | 2015 | 64.8 per 100,000 | 42.2 per 100,000 |
| Uganda | East and Southern Africa | Low | Two-Dose | 2015 | 53.8 per 100,000 | 40.6 per 100,000 |

of emerging insights [18,19]. Five study team members (SR, MDW, ENR, IS, CG) based in the United States conducted English- or French-language in-depth interviews (lasting 30-60 minutes) via secure teleconference platform, aided by a semi-structured guide—allowing the dialogue to proceed iteratively in pursuit of emergent concepts and themes (see S1 Checklist). Interviewers were trained in qualitative research methods and had familiarity with HPVV and cervical cancer control programmes, immunization services in LMICs, and global vaccine policy. The interview guide elicited stakeholders' perspectives on the following domains and topics: informants' professional role and experience with country immunization programmes; perceptions of government prioritization of HPV immunization; stakeholders involved in single-dose HPVV decision-making; and the sequence of events concluding in HPVV schedule decision-making at country level. Informants did not receive any compensation for completing an in-depth interview.

## Analysis

Interviews were audio-recorded, professionally transcribed, and inspected for transcription quality and fidelity to the recorded dialogue. We implemented a multi-step, team-based approach to qualitative data analysis, beginning with close, line-by-line reading of interview transcripts. Following initial familiarization with the data corpus, we conducted purposive text analysis [20] to: (1) characterize the sequence of events resulting in a decision being rendered on the HPVV schedule; (2) enumerate actors involved at each stage of HPVV schedule decision-making; and (3) identify key forces (e.g., institutions, policies, contextual factors) facilitating or impeding momentum along the HPVV decision-making pathway(s). Compared to classical thematic analysis approaches, which involve systematic condensation and reassembly of textual data to identify salient themes [21], purposive text analysis entails close inspection of textual data to derive meaning or inferences about phenomena defined *a priori* [20]. In

our application of purposive text analysis, we sought to extract cause-and-effect relationships (or temporally situated sequences) between salient events articulated from stakeholders' narratives of HPVV decision-making at country level.

After re-reading interview transcripts and generating analytic memos summarizing initial impressions of similarities and differences in country-level HPVV decision-making processes, as well as actors involved at each phase of decision-making, we developed a data abstraction matrix to manually sort text segments related to HPVV decision-making into antecedents (i.e., causes) and consequences (i.e., effects). The first author (SR) then reassembled coded text segments in sequential order, identifying a chronology of events—and actors involved during each step—in the HPVV single-dose decision-making continuum. After identifying a preliminary sequence of activities/events within each country (vertical analysis), the first author inspected data elements across cases (horizontal analysis) to discern patterns and heterogeneities in HPVV single-dose decision-making pathways and process across the 19 LMICs included [22,23]. To enhance the credibility and confirmability of findings [24], emerging results were continuously refined through discussions amongst the study team.

## Ethics statement

The study protocol was reviewed and deemed exempt from further human subjects research oversight by the Johns Hopkins University Bloomberg School of Public Health Institutional Review Board (Baltimore, Maryland). All informants were provided with a study information sheet outlining the study objectives and reaffirming their voluntary participation in an interview. Because the study received human subjects research exemption from the Johns Hopkins University Bloomberg School of Public Health Institutional Review Board, and no personally identifiable information was obtained from key informants, informed consent was not obtained prior to interview conduct.

## Results

We conducted interviews with 66 national immunization stakeholders (MoH/EPI or NITAG/TWG members [$n = 26$], technical assistance partners or civil society organizations [$n = 22$], multilateral organizations [$n = 18$]) from 19 LMICs, 11 of which had introduced HPVV before the publication of the WHO position paper on single-dose HPVV (Burkina Faso [$n = 5$], Cambodia [$n = 3$], Cote d'Ivoire [$n = 4$], Eswatini [$n = 3$], Ethiopia [$n = 4$], Indonesia [$n = 3$], Kenya [$n = 5$], Laos [$n = 3$], Philippines [$n = 4$], Tanzania [$n = 5$], Uganda [$n = 2$]), and 8 countries were classified as new HPVV introductions (Bangladesh [$n = 3$], Cameroon [$n = 5$], Ghana [$n = 4$], India [$n = 2$], Madagascar [$n = 1$], Nepal [$n = 3$], Nigeria [$n = 3$], Pakistan [$n = 4$]). Ten of these countries had adopted or switched to single-dose HPVV at the time of data collection (Bangladesh, Burkina Faso, Cambodia, Cameroon, Cote d'Ivoire, Ethiopia, India, Laos, Nigeria, Tanzania).

Interviews highlighted the heterogeneity of actors shaping country-level dialogues related to HPVV schedules, as well as the sequence of activities/events concluding in an HPVV schedule decision. In what follows, we describe the stakeholders and agencies involved in HPVV single-dose decision-making and the multi-phased process to arrive at a decision related to the immunization schedule in 19 LMICs.

### Actors and agencies involved in HPVV schedule decision-making

Immunization stakeholder narratives underscored the presence of the MoH/EPI, NITAG or equivalent TWG, and the WHO in country-level dialogues and deliberations related to the HPVV schedule. Stakeholders described the MoH's role as the primary decision-maker

in-country of the HPVV schedule while emphasizing the NITAG's function of collating and synthesizing evidence as critical to guiding the MoH's recommendation.

*"Together with the NITAG, they [the MoH] will review scientific evidence…to ensure that they [HPVV schedule decisions] will not have short or long-term negative impacts and will have more positive impact on the population and expand opportunities to other populations as well. But the final decision will be made by the Ministry of Health."* –**Laos**

*"It is the Ministry of Health ultimately, but the Ministry must seek advice from these all committees, starting with the TWG and the NITAG mostly. The NITAG is the one who gives the final recommendation. If the NITAG does not approve, then it is difficult for the Ministry to go ahead with any change in the immunization schedule."* –**Tanzania**

Across countries, immunization stakeholders emphasized the centrality of the NITAG to HPVV schedule decision-making, given their mandate to consolidate evidence and generate a recommendation to the MoH. In the Philippines, for example, decision-making related to the HPVV schedule had stalled because the NITAG had been disbanded; the NITAG would need to be reconstituted before an official recommendation on the HPVV schedule could be issued to the MoH.

*"There is a process of reconstituting the NITAG, but for the moment, the NITAG is not there. It has to be [decided] after the NITAG [is reconstituted]."* –**Philippines**

In some LMICs, peripheral stakeholders like multilateral organizations (e.g., United Nations agencies), in-country technical assistance partners, other country line ministries (e.g., Ministry of Finance, Ministry of Education), professional societies like the pediatric and gynecological associations, and the Interagency Coordination Committee (ICC) featured more prominently in HPVV schedule decision-making. For example, core/expanded immunization partners supported NITAG activities by compiling evidence dossiers for single-dose HPVV or related information-sharing that guided NITAG deliberations. Furthermore, in other settings (e.g., Cameroon), MoH and NITAGs mobilized professional associations and other country line ministries to contribute expertise to the HPVV delivery strategies that could feasibly and acceptably accommodate a single-dose HPVV schedule.

*"We decided to review our [HPVV] strategy and see what could be done better. To do this, we decided to work closely and organize working sessions with actors from religious authorities—Christians, Muslims, and their leaders. We also worked with expert groups like the Cameroon Association of Obstetricians and Gynecologists and the Association of Pediatricians…We worked with other Ministries like the Ministry of Basic Education."* –**Cameroon**

In other countries with more decentralized systems of governance, regional and provincial governments were engaged and influenced NITAG deliberations and MoH recommendations, despite the national scope of these two bodies. In Pakistan, for instance, stakeholders explained that provincial governments ultimately have decision-making authority over the immunization schedules for their respective jurisdictions, given their oversight over the financing of immunization programmes and services.

*"The provincial government would convey their messages or their decision to the national government…It would be a consultative thing, but the decision would be made by the*

*provincial governments, and then their decision would be conveyed to the national government, and the national government would further convey it to Gavi and other stakeholders."* –**Pakistan**

Thus, in these circumstances, the NITAGs deliver a final recommendation but not without extensive consultations with subnational governments—reaffirming the pivotal roles of bodies peripheral to the MoH and NITAG in shaping HPVV schedule deliberations and decision-making.

### Sequence of HPVV schedule decision-making at country level

Fig 2 illustrates a sequential pathway of events concluding in a rendered decision on the HPVV schedule, derived from purposive text analysis of immunization stakeholder narratives. The decision-making sequence typically begins with the WHO and SAGE issuing new recommendations or guidelines related to the HPVV schedule, which are subsequently disseminated at country level to MoH and EPI. From there, NITAGs are convened at the MoH's request to review the revised immunization guidelines, evidence underpinning the revised immunization schedule recommendations, and issue a schedule-related recommendation to the MoH, typically after multiple discussion and workshop-style meetings. After the NITAG delivers a recommendation to the MoH, the MoH could either reject or approve the recommendation by ratifying an HPVV schedule in alignment with the NITAG's recommendation. Lastly, the ICC reviews the Ministry's recommendation and provides final authorization before a change to the immunization schedule is communicated to Gavi.

1. **New evidence emerges and informs global HPVV guidelines:** The updated SAGE recommendations and accompanying WHO position paper in 2022 initially triggered national immunization programmes to revisit their current or planned HPVV schedules. One stakeholder from Eswatini compared the WHO position paper to "*Bible verses,*" reaffirming how the NITAG's role is primarily focused on revalidating the evidence informing WHO

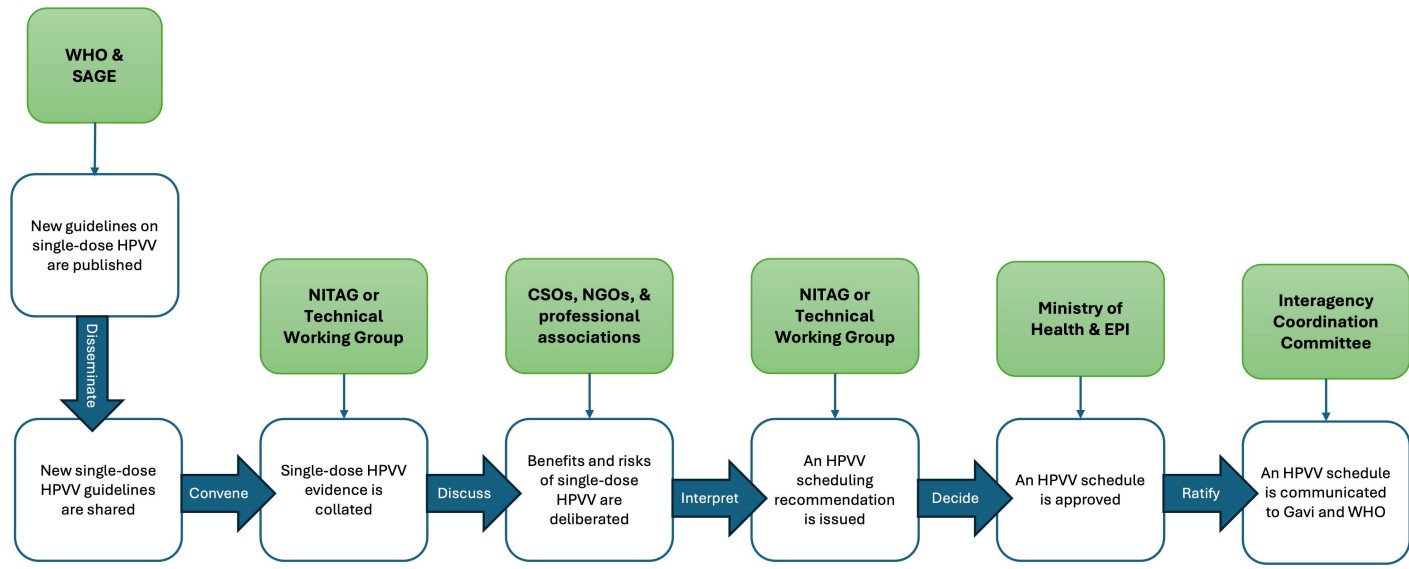

**Fig 2. Decision-making pathway and process to adopting or switching to the single-dose HPV vaccination schedule in 19 low- and middle-income countries.**

guidelines and operationalizing revised WHO recommendations in country context, rather than calling the HPVV schedule recommendations writ large into question.

2. **Revised schedule recommendations are communicated to MoH and EPI:** Once the updated SAGE recommendation and corresponding WHO position paper on single-dose HPVV were publicized, WHO proactively disseminates the revised guidelines to national immunization programmes through various modalities (i.e., bilateral memoranda, public-facing webinars). From there, the MoH then convenes and tasks the NITAG with consolidating evidence and contextualizing the new guidelines within their national contexts.

   *"The SAGE and WHO position papers guide us on requirements, and once that is out, then we have to hold sensitization meetings with our partners and our stakeholders, so we inform others of the WHO's position on...single-dose [HPVV]."* – **Kenya**

3. **Collation, deliberation, and assessment of HPVV single-dose evidence:** Once the NITAG or equivalent TWG begins collating and synthesizing the evidence to support an HPVV schedule decision, they typically convened multiple times as a group to discuss the breadth and quality of evidence underpinning the single-dose recommendation, as well as the various social, financial, and health-related implications of adopting or rejecting the single-dose HPVV schedule. In some circumstances, the NITAGs consulted technical assistance partners, professional associations, and other country line ministries to help synthesize the emerging evidence or communicate information about implementation lessons learned from countries that had adopted a single-dose HPVV schedule.

   *"We have the full NITAG committee, and then we have what we call technical working groups. We had a technical working group that was specifically looking into the information regarding the single dose, and this [group] met around three times. And then they developed a draft report, which they presented to the full committee for consideration. I think the full committee met twice, and the technical working group met about three or four times."* – **Uganda**

   *"They [NITAGs] have to look at [vaccine] efficacy and effectiveness, and this is why we are bringing in the consultant to support us on that. It's the issue of the burden of the disease... We're also looking at the priority groups that are going to benefit and the cost-effectiveness in terms of the disease, daily adjusted life years, and the vaccine costs."* – **Kenya**

4. **Finalization and ratification of an HPVV schedule recommendation:** Once the NITAG accumulates sufficient evidence and reaches internal consensus among its members, typically over 3-4 convenings, they would then issue an official HPVV schedule recommendation to the MoH, who would respond affirmatively to the recommendation or request additional evidence/information prior to ratifying the NITAG's recommendation. The ICC then formalizes the MoH's decision by convening a larger body of stakeholders beyond those represented in the NITAG, lending additional visibility to the decision-making process. In some circumstances, a recommendation might be privately endorsed but not formally ratified by the MoH, particularly in situations when the financial implications of the HPVV schedule recommendation remain uncertain.

   *"At the end, the NITAG recommended to the Ministry of Health to introduce the vaccine... We then took the recommendation to the Interagency Coordination Committee, which is a*

*committee presided by the Minister of Health himself, which brings all the immunization stakeholders together including United Nations agencies, civil society organizations, and the public administration, education, and economic ministries to see if what has been recommended by the expert group could be validated. After we validated the recommendation of the expert group at that level, we did what we call a national vaccine introduction plan."* –**Burkina Faso**

*"The final decision is made by the Minister, who is not a NITAG member, but the NITAG makes a recommendation to him, so he will look at the fiscal space—whether the money is available or not. If money is not available, the recommendation will be delayed until money becomes available, and that is what we are waiting for now. At the time we made the decision [to introduce HPVV], the two-dose vaccine was available, but now with a single dose, the decision is easier because the money required will now be less, and at this stage, that is the recommendation to the Minister."* –**Ghana**

## Discussion

Across 19 LMICs eligible for Gavi co-financing, NITAGs emerged as pivotal actors in decision-making processes related to the HPVV schedule. These multidisciplinary technical advisory groups, alongside the MoH, exhibit significant influence in the timing and pace of national decision-making related to HPVV schedules. Prior research has generated mixed evidence on the perceived influence of NITAGs on immunization decision-making, with some studies deemphasizing the authority of NITAGs in immunization-related decision-making relative to other policymaking bodies (e.g., WHO, MoH/EPI) [16,25], while others have underscored the supremacy of NITAGs in facilitating new vaccine introductions and product switches through evidence collation/synthesis [26]. Our findings, nevertheless, demonstrate the instrumental role of NITAGs in interpreting and operationalizing global immunization guidelines within their national context—reaffirming the importance of investments to sustain convened technical bodies like NITAGs to facilitate timely, evidence-informed vaccine decision-making.

Stakeholder interviews underscored the considerable diversity of actors and institutions influencing the velocity and trajectory of HPVV schedule decision-making in Gavi-eligible LMICs. Beyond MoH/EPI and other country line ministries, a constellation of stakeholders (e.g., professional associations, civil society organizations) provided testimony and contributed to national dialogues related to single-dose HPVV. This underscores the complexity of national immunization decision-making and reaffirms the need for timely evidence dissemination that empowers heterogenous stakeholders to participate inclusively in and contribute to HPVV schedule decision-making. Despite evidenced-grounded HPVV schedule decision-making observed across 19 LMICs included in our study, other contextual factors (e.g., volatility of political commitments to HPVV, financial uncertainties to subsidizing the HPVV programme) impacted the timing and trajectory of HPVV schedule decision-making. New vaccine introductions for other antigens like pneumococcal conjugate vaccine (PCV) demonstrate the susceptibility of vaccine implementation to the pace of national decision-making, especially in contexts where political and financial authority is concentrated in decentralized policymaking bodies (e.g., subnational governments) that can attenuate the influence of federalized decision-makers like NITAGs and MoH/EPI [27,28]. Decision-making related to immunization schedules for PCV [27,28], rotavirus vaccines [26], and other pediatric antigens [29] has been stalled by similar contextual forces in LMICs. This underscores the need for comprehensive, holistic decision-making supports that consider not only evidence availability/quality but also practical constraints and systemic realities in the implementation of immunization programmes. Furthermore, the dynamic nature of evidence generation (and

the multilevel drivers of their dissemination and translation to use) in the immunization space requires highly responsive, resilient decision-making structures at national level. To accomplish this, LMICs should invest in strengthening NITAGs or equivalent technical bodies by ensuring they have the requisite expertise, resources, and independence to rapidly synthesize scientific evidence and situate evolving global immunization guidelines in national context [30,31]. Additionally, fostering independence in evidence collation and synthesis through expanded local capacity in various technical areas (e.g., cost-effectiveness, modelling) can facilitate country ownership of HPVV schedule decision-making [32].

Our findings are subject to several limitations. First, given the evolving nature of HPVV schedule decision-making, perspectives shared by stakeholders at a singular point in time may become outdated shortly after their communication during in-depth interviews. Second, the sensitivity of HPVV decision-making, especially in countries that had yet rendered a decision on the immunization schedule, might have demotivated stakeholders from sharing or divulging specific details or information during the interviews; the professional networks from which stakeholders were sampled may have further amplified sensitivities surrounding these interviews and prompted non-disclosure of specific HPVV pathways/processes by participants. Third, in-depth interviews with immunization stakeholders emphasized country-led pathways/processes for determining vaccination schedules, likely overlooking political determinants of these processes (e.g., influence of pharmaceutical companies and lobbying groups). Given that immunization schedule decision-making is both a scientific and political process, future studies should explicitly characterize the influence of political actors on pathways/processes to HPVV schedule-related decision-making in LMICs. Lastly, insights gleaned from the present study represent perspectives from national immunization stakeholders in 19 LMICs; findings, thus, may not be transferrable to the decision-making pathways/processes of countries that are not represented in the present study, including high-income countries or other middle-income countries that are ineligible for Gavi co-financing for national HPVV introductions and coverage improvement efforts.

## Conclusions

This is among the first studies to document the decision-making processes and pathways to adopting or switching to a single-dose HPVV schedule in 19 LMICs, providing a blueprint for other LMICs who may introduce HPVV in the future with Gavi support. We found that functional NITAGs are imperative to HPVV schedule decision-making and exert considerably influence over the timing and pace of adopting or switching to single-dose HPVV schedule. While HPVV programmes are in an era of revitalization (following a period of dormancy during COVID-19) [33], and cervical cancer elimination is a burgeoning global priority [4], LMICs must invest adequate financial and human resources to constituting and sustaining their NITAGs, who are among the more critical actors to facilitating evidenced-based decision-making for HPVV programmes.

## Supplementary information

**S1 Checklist. Inclusivity in global health questionnaire.**
(DOCX)

## Acknowledgments

We extend our gratitude to the national immunization stakeholders who shared their time and insights, without whom this work would not be possible. We also acknowledge representatives from the HAPPI Consortium (JSI Research & Training Institute, Inc., Clinton Health Access

Initiative, Jhpiego, PATH); the Coalition to Strengthen the HPV Immunization Community; and the Gavi HPV vaccine sub-team who supported protocol development, stakeholder mapping, informant recruitment, and results interpretation.

## Author contributions

**Conceptualization:** Rupali J. Limaye, Joseph Gregory Rosen.

**Formal analysis:** Sunny Roy, Megan D. Wysong, Erica N. Rosser, Ishani Sheth, Casey Geddes, Joseph Gregory Rosen.

**Funding acquisition:** Rupali J. Limaye, Joseph Gregory Rosen.

**Investigation:** Sunny Roy, Megan D. Wysong, Erica N. Rosser, Ishani Sheth, Casey Geddes.

**Project administration:** Megan D. Wysong.

**Supervision:** Rupali J. Limaye, Joseph Gregory Rosen.

**Writing – original draft:** Sunny Roy.

**Writing – review & editing:** Megan D. Wysong, Erica N. Rosser, Ishani Sheth, Casey Geddes, Rupali J. Limaye, Joseph Gregory Rosen.

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
