## [Decision Letter · Decision Letter 0]

17 Dec 2024

PGPH-D-24-01855

Pathways and Processes to Adopting or Switching to a Single-Dose HPV Vaccination Schedule in Low- and Middle-Income Countries: A Qualitative Study

Dear Dr. Rosen,

Thank you for submitting your manuscript to PLOS Global Public Health. After careful consideration, we feel that it has merit but does not fully meet PLOS Global Public Health’s publication criteria as it currently stands. Therefore, we invite you to submit a revised version of the manuscript that addresses the points raised during the review process.

Comments from reviewers are mostly positive. Please do respond, and make a note if you do not agree with any comments.

We look forward to receiving your revised manuscript.

Kind regards,

Abram L. Wagner, PhD, MPH

Academic Editor

Journal Requirements:

2. Please provide separate figure files in .tif or .eps format.

Additional Editor Comments (if provided):

Reviewers' comments:

Reviewer's Responses to Questions

**Comments to the Author**

1. Does this manuscript meet PLOS Global Public Health’s publication criteria ? Is the manuscript technically sound, and do the data support the conclusions? The manuscript must describe methodologically and ethically rigorous research with conclusions that are appropriately drawn based on the data presented.

Reviewer #1: Yes

Reviewer #2: Yes

2. Has the statistical analysis been performed appropriately and rigorously?

Reviewer #1: N/A

Reviewer #2: N/A

3. Have the authors made all data underlying the findings in their manuscript fully available (please refer to the Data Availability Statement at the start of the manuscript PDF file)?

Reviewer #1: No

Reviewer #2: No

4. Is the manuscript presented in an intelligible fashion and written in standard English?

Reviewer #1: Yes

Reviewer #2: Yes

5. Review Comments to the Author

Reviewer #1: The manuscript reports the findings of a study that aims to characterize the sequence of events, as well as actors/institutions involved, that facilitate or impede country-level decision-making related to the HPV vaccine schedule. With the new recommendations for countries to consider using single dose HPV vaccine by WHO and its SAGE and the high burden of cervical cancer in low and middle income countries, this is an important and valuable study to understand the influencing factors in countries being able to consider and introduce the new schedule.

The authors have systematically described the methods and process of analysis in good depth and clarity. Clarification on the ethical considerations are also made clear. Data is not made available but the authors have responded to the PLOS question on data availability regarding it being sensitive information and that they would be shared upon request in a de-identifying manner.

Two things that the authors should consider and further clarify in their manuscript -

1. Although through interviews the study has been able to demonstrate the processes, key actors and agencies, and factors influencing the decision making, the study has limitations in not having explored the underlying political forces in operation which may remain unsaid in the interviews. The influence of lobby groups, pharmaceutical industry and other powerful agencies have over even the so-called “scientific” body or its recommendations. The idea that these processes are not merely scientific but also deeply political. Perhaps the authors can clarify this as one of their study limitations. The role of peripheral bodies are explored to a certain extent and further discussed in the discussion section but by design, the study hasn’t explored the political root causes/processes.

2. The current study team’s organization and study’s funding organization are key players in the networks through which the participants were recruited. This would have additional risk of participants withholding sensitive information and is a limitation that can be mentioned.

Reviewer #2: This manuscript examines immunization stakeholders’ roles in low- and middle-income countries (LMICs) to better understand how crucial the stakeholder groups’ HPV vaccine (HPVV) recommendations are after the WHO revised the HPVV schedule to support single dose vaccinations. The authors effectively outlined the need for this type of qualitative research by providing the reader with a clear understanding of why HPVVs are important, the benefit of single dose schedules, and how stakeholder groups (i. e. NITAGs) influence formal vaccine schedules.

Having 66 participants interviewed allows for diverse representation. However, the authors should include the specific professional backgrounds/positions of the participants and the number of participants from each country, so the reader can note which countries had more interview participants. The methods section refers to the interview guide that included the participants’ professional roles, therefore these data should be provided in the manuscript to display stakeholder/participant qualifications (doctors, epidemiologists, etc.). Providing tables with where each participant was from and their positions would allow the reader to determine the diversity (or lack of) of the participant population. Providing these data should demonstrate reliability. Lastly, table 1 should be reviewed for consistent language in the “income status” column.

Overall, this manuscript will be a valuable contribution to existing STI prevention and public health/policy literature by outlining how NITAGs’ recommendations determine LMICs’ HPVV schedules and what resources (i.e. funding) impact those recommendations. This paper provides a sound analysis of NITAGs in LMICs and how important their role in global health is.

6. PLOS authors have the option to publish the peer review history of their article (what does this mean? ). If published, this will include your full peer review and any attached files.

**Do you want your identity to be public for this peer review?** For information about this choice, including consent withdrawal, please see our Privacy Policy .

Reviewer #1: No

Reviewer #2: No

---

## [Editor Report · Decision Letter 1]

16 Jan 2025

Pathways and Processes to Adopting or Switching to a Single-Dose HPV Vaccination Schedule in Low- and Middle-Income Countries: A Qualitative Study

PGPH-D-24-01855R1

Dear Mr. Rosen,

We are pleased to inform you that your manuscript 'Pathways and Processes to Adopting or Switching to a Single-Dose HPV Vaccination Schedule in Low- and Middle-Income Countries: A Qualitative Study' has been provisionally accepted for publication in PLOS Global Public Health.

Best regards,

Abram L. Wagner, PhD, MPH

Academic Editor